# Theanine, Antistress Amino Acid in Tea Leaves, Causes Hippocampal Metabolic Changes and Antidepressant Effects in Stress-Loaded Mice

**DOI:** 10.3390/ijms22010193

**Published:** 2020-12-28

**Authors:** Keiko Unno, Yoshio Muguruma, Koichi Inoue, Tomokazu Konishi, Kyoko Taguchi, Sanae Hasegawa-Ishii, Atsuyoshi Shimada, Yoriyuki Nakamura

**Affiliations:** 1Tea Science Center, University of Shizuoka, 52-1 Yada, Suruga-ku, Shizuoka 422-8526, Japan; gp1719@u-shizuoka-ken.ac.jp (K.T.); yori.naka222@u-shizuoka-ken.ac.jp (Y.N.); 2Graduate School of Pharmaceutical Sciences, Ritsumeikan University, 1-1-1 Nojihigashi, Kusatsu, Shiga 525-8577, Japan; ph0063hv@ed.ritsumei.ac.jp (Y.M.); kinoue@fc.ritsumei.ac.jp (K.I.); 3Faculty of Bioresource Sciences, Akita Prefectural University, Shimoshinjo Nakano, Akita 010-0195, Japan; konishi@akita-pu.ac.jp; 4Faculty of Health Sciences, Kyorin University, 5-4-1 Shimorenjaku, Mitaka, Tokyo 181-8612, Japan; sanae_ishii@ks.kyorin-u.ac.jp (S.H.-I.); ats7@ks.kyorin-u.ac.jp (A.S.)

**Keywords:** carnosine, histamine, kynurenine, ornithine, SAMP10, stress, theanine

## Abstract

By comprehensively measuring changes in metabolites in the hippocampus of stress-loaded mice, we investigated the reasons for stress vulnerability and the effect of theanine, i.e., an abundant amino acid in tea leaves, on the metabolism. Stress sensitivity was higher in senescence-accelerated mouse prone 10 (SAMP10) mice than in normal ddY mice when these mice were loaded with stress on the basis of territorial consciousness in males. Group housing was used as the low-stress condition reference. Among the statistically altered metabolites, depression-related kynurenine and excitability-related histamine were significantly higher in SAMP10 mice than in ddY mice. In contrast, carnosine, which has antidepressant-like activity, and ornithine, which has antistress effects, were significantly lower in SAMP10 mice than in ddY mice. The ingestion of theanine, an excellent antistress amino acid, modulated the levels of kynurenine, histamine, and carnosine only in the stress-loaded SAMP10 mice and not in the group-housing mice. Depression-like behavior was suppressed in mice that had ingested theanine only under stress loading. Taken together, changes in these metabolites, such as kynurenine, histamine, carnosine, and ornithine, were suggested to be associated with the stress vulnerability and depression-like behavior of stressed SAMP10 mice. It was also shown that theanine action appears in the metabolism of mice only under stress loading.

## 1. Introduction

Under similar circumstances, the stress response is heterogeneous in both humans and experimental animals. There is a difference in susceptibility to stress, that is, some people develop stress-related disorders such as depression and post-traumatic stress disorder, while others do not. To elucidate the difference in stress susceptibility, senescence-accelerated mouse prone 10 (SAMP10) mice are suitable as a stress-vulnerable model. SAMP10 mice are known to display age-related characteristic brain atrophy and depression-like behavior [1,2]. In addition, when SAMP10 mice were loaded with stress on the basis of territorial consciousness in males, stressed SAMP10 mice showed accelerated brain atrophy, cognitive dysfunction, and lifespan shortening [3]. The stress was based on territorial consciousness as follows: After two male mice were housed in a partitioned cage for one month to establish territorial consciousness (single housing), the partition was removed to expose the mice to psychosocial confrontational stress, and the two mice subsequently cohabited in the same cage (confrontational housing). Volumetric brain changes induced by psychosocial stress have been observed one month after confrontational housing in SAMP10 mice, and atrophy continued thereafter [4]. Cerebral atrophy has also been observed in ddY mice, a strain that ages normally, but this was temporary. Significant adrenal hypertrophy, a typical stress response, has been observed at least one week after confrontational housing in ddY mice [5], and therefore it was considered that SAMP10 and ddY mice felt similar psychosocial stress by confrontational housing. However, stress due to confrontational housing may not last long in ddY mice. We investigated changes in gene expression in the hippocampus on the third day of stress loading, and clarified the difference between SAMP10 and ddY mice, that is, the expression levels of neuronal PAS domain protein 4 (*Npas4*) and lipocalin 2 (*Lcn2*) were involved in the brain atrophy and stress vulnerability of SAMP10 ice, and the changed expression of *Npas4* and *Lcn2* was prevented by theanine ingestion [4]. Theanine is a nonprotein amino acid that exists almost exclusively in tea (*Camellia sinensis* L.) leaves. Theanine intake suppresses psychosocial stress [6]. Indeed, adrenal hypertrophy, a typical stress response, has been significantly suppressed in mice that have ingested theanine even under stress loading [3,5]. In addition, brain atrophy due to chronic stress has been significantly suppressed in SAMP10 and ddY mice that ingested theanine [3]. Npas4 is a transcription factor that plays a role in the development of inhibitory synapses [7]. Lcn2, which is primarily secreted by reactive astrocytes, directly induces neuronal damage and amplifies neurotoxic inflammation under many brain conditions [8]. These early changes in gene expression are considered to be one reason for the stress vulnerability of SAMP10 mice. There is a need to investigate how these changes in gene expression subsequently affect brain metabolism. We focused on the metabolites that have been used to evaluate the diversity of amine-mediated metabolic patterns and pathways that are a confirmed diagnosis based on the pathophysiology of the brain in Alzheimer’s disease [9].

The hippocampus is a stress-vulnerable tissue in the brain. Neurogenesis in the hippocampus occurs throughout the life of a wide range of animal species, and it could be associated with hippocampus-dependent learning and memory [10,11,12]. However, restrained chronic stress has been shown to significantly decrease hippocampal volume and impair hippocampal neurogenesis in mice [13]. Hippocampal neurogenesis reportedly plays an important role in the regulation of the inhibitory circuitry of the hippocampus [14]. In addition, the maintenance of a balance between inhibitory and excitatory elements in the brain is believed to be important for synaptic plasticity and cognitive function [15,16], and the regulation of inhibitory neuronal activation may be especially important in the hippocampus during chronic stress [17,18,19,20]. Thus, we focused on the hippocampus in both SAMP10 and ddY mice that were housed confrontationally. Group-housing mice were used as a model for low-stress conditions. 

To examine the reason for the different stress-sensitivity levels between SAMP10 and ddY mice, brain metabolites have been compared between SAMP10 and ddY mice. In this study, hippocampal metabolites were analyzed by comparing confrontational and group-housing SAMP10 and ddY mice that ingested theanine water or control water. From the obtained results, some factors related to depressive behavior were found. Therefore, the effects of confrontational housing and theanine ingestion on depression-like behavior of SAMP10 mice were examined. In addition, the expression of several enzymes involved in the synthesis of these metabolites was measured.

## 2. Results

### 2.1. Effects of Theanine Ingestion on Brain Metabolites in Senescence-Accelerated Mouse Prone 10 (SAMP10) and ddY Mice Stressed by Confrontational Housing 

The SAMP10 and ddY mice were divided into two groups, i.e., confrontational and group-housing, respectively. These mice were further divided into two groups that ingested theanine or control water. For confrontational housing, two mice were housed in a partitioned cage for one month to establish territorial consciousness (single housing). Then, the partition was removed to expose the mice to confrontational stress, and the two mice subsequently cohabited in the same cage (confrontational housing) for one month (Figure 1). 

The effects of theanine intake on metabolites in the hippocampus of mice stressed by confrontational housing were analyzed by principal component (PC) analysis [21]. The group-housing mice were used as the reference of a low-stress condition. Both PC1 and PC2 displayed differences between SAMP10 and ddY mice, and they both indicated the difference of housing conditions and theanine treatment (Table 1). Metabolites were analyzed in both SAMP10 and ddY mice together. Thirty-eight metabolites that were positive in ANOVA were analyzed by PCA (Table 2). The PC for samples presented characteristics of the groups on axes of PC1 and PC2. First, PC1 detected the difference between SAMP10 and ddY mice (Table 1). On both axes, conditions and treatment were separated differently wfor SAMP10 and ddY mice, which showed that condition and treatment had different effects on ddY and SAMP10 mice. On the one hand, in ddY mice, there was a significant difference in conditions on the PC1 axis, and there was not much change on the PC2 axis. On the other hand, the SAMP10 mice required theanine, as well as a confrontation to increase PC1, but the direction of change was the same as that of the ddY mice. This combination also showed a significant difference on the PC2 axis.

PC for metabolites is inextricably linked to PC for samples in a mathematical sense. Confrontation alone decreases in ddY mice, but in SAMP10, the substances that needed more theanine were kynurenine (Kyn) and histamine. In addition, it was 5-methoxytryptamine that decreased only in SAMP10 mice with the combination of confrontation and theanine (Table 2).

Some metabolites with negative PC1 scores tend to be higher in SAMP10 mice (Table 2 and Figure 2). The levels of Kyn in the group-housing SAMP10 mice were significantly higher than those of the ddY mice that had ingested theanine or not (Figure 2a). Similarly, the levels of histamine, 5-methoxytryptamine, 2,4-diamonobutyric acid, and histidinol were higher in the SAMP10 mice than in ddY mice (Figure 2b–e). However, these levels were significantly lower in the SAMP10 mice that had ingested theanine under confrontational housing than in those under group housing. Kyn and 5-metoxytryptamine are tryptophan (Trp) metabolites, which both present strong signals in PCA (Table 2).

In contrast, guanosine monophosphate (GMP), which showed a positive PC1, showed significantly higher values for ddY mice in group and confrontational housing than those of the SAMP10 mice, when theanine had not been ingested (Figure 2f). The level of carnosine was significantly higher in the ddY mice than in the SAMP10 mice with and without theanine intake (Figure 2g). Putrescine (Put) was significantly higher in the ddY mice than theSAMP10 mice for the confrontational-housing mice that ingested no theanine (Figure 2h). Ornithine (Orn) was significantly higher in the ddY mice than in the SAMP10 mice for the group and confrontational housing mice that had ingested no theanine (Figure 2i). In addition, theanine ingestion increased the level of Orn in SAMP10 (Figure 2i). Adenosine was significantly higher in the ddY mice than in the SAMP10 mice for the group and confrontational housing mice that had ingested theanine or not (Figure 2j).

### 2.2. Effect of Theanine Ingestion on Depression-Like Behavior

To evaluate depression-like behavior in mice, the tail suspension test is used widely for preclinical screening of antidepressants. Mice are considered to present immobility as depression-like behavior after they are subjected to inescapable stress and failure in efforts to free themselves. As a result of determining the depressive behavior of the SAMP10 mice by the tail-suspension test, the effect of theanine intake was not observed in the group-housing mice, but immobility time was reduced significantly by theanine intake in mice under confrontational housing (Figure 3). Similarly, in ddY mice, theanine intake significantly reduced immobility time under confrontational housing. Reduced immobility time meant that depression-like behavior was improved.

It was not possible to compare SAMP10 and ddY mice because the optimal observation time depended on the mouse strain, but it was possible to compare the effects of confrontational housing on depression-like behavior within the same strain. There was no significant difference between group and confrontational housing in the control mice, and the presence or absence of theanine intake had no effect on immobility time. However, immobility time was shortened only in mice that ingested theanine under confrontational housing. 

### 2.3. Effect of Theanine Ingestion on the Levels of Indoleamine/Tryptophan-2,3-dioxygenase, Arginase and Histidine Decarboxylase

Since Kyn is produced from Trp via indoleamine/tryptophan-2,3-dioxygenase (IDO, TDO), these expression levels in the hippocampus were compared between SAMP10 and ddY mice (Figure 4a,b). The expression of IDO was significantly higher in the control SAMP10 mice under confrontational housing than that of the group-housing SAMP10 mice and that of the control ddY mice under confrontational housing. The IDO levels were also significantly higher in the SAMP10 mice than in ddY mice that ingested theanine under confrontational housing. The level of TDO in the control SAMP10 mice was lower than the control ddY mice. The levels of TDO were not changed significantly in the SAMP10 and ddY mice when they ingested theanine. These results indicated that the expression of IDO was increased in the SAMP10 mice but not in the ddY mice under confrontational housing.

Ornithine is produced from arginine via arginase. The expression level in the hippocampus was compared (Figure 4c) and the expression level was significantly increased in SAMP10 mice that ingested theanine, but was lowered by confrontational housing. The levels of arginase in ddY mice were not changed. 

Histamine is synthesized from histidine via histidine decarboxylase (HDC), in which mRNA is expressed exclusively in the posterior hypothalamus. The expression levels in the hypothalamus were compared (Figure 4d). The expression of HDC in the hypothalamus was higher in ddY mice that ingested theanine under group housing than in the SAMP10 mice. The level in ddY mice was significantly suppressed by confrontational housing.

## 3. Discussion

Changes in hippocampal metabolites were compared among mice with different stress susceptibilities. The levels of Kyn, histamine, 5-metoxytryptamine, 2,4-diaminobutyric acid, and histidinol were significantly higher in the SAMP10 mice than in the ddY mice. However, these levels were suppressed in the SAMP10 mice that had ingested theanine under confrontational housing. The Kyn level did not decrease in the group-housing SAMP10 mice that had ingested theanine, suggesting that theanine began to act only when some metabolism changes occurred due to stress loading. Kyn has been reported to increase in chronically stressed rats [22]. In addition, the Kyn pathway plays a key role in depression-like behavior in mice [23,24]. Kyn levels were lower in the ddY mice than in the SAMP10 mice, but even lower when the ddY mice ingested theanine under confrontational housing. Similarly, shortening of immobility time was observed in both the SAMP10 and ddY mice that ingested theanine under confrontational housing. 

Kyn is produced from Trp via IDO/TDO and TDO is an enzyme present in the liver [25]. The effects of stress loading and theanine ingestion on IDO and TDO expression levels were investigated. The level of IDO was significantly higher in the SAMP10 mice than in the ddY mice under confrontational housing, while the levels were not changed among the SAMP10 and ddY mice under group housing with or without theanine ingestion (Figure 4a). The high expression level of IDO in the SAMP10 mice under confrontational housing may contribute to the high amount of Kyn in the SAMP10 mice. However, the increased level of Kyn level in the group-housing SAMP10 mice that ingested theanine could not be explained by the expression levels of IDO and TDO alone.

Carnosine presents at a high concentration in the brain and has been reported to have antidepressant-like activity [26,27]. Carnosine levels were significantly lower in the SAMP10 mice than in the ddY mice, but the levels were significantly increased in the SAMP10 mice that had ingested theanine under confrontational housing. Depression-like behavior was suppressed in the SAMP10 and ddY mice that had ingested theanine under confrontational housing, where Kyn was suppressed, and carnosine was increased. The suppression of depression-like behavior only in mice that ingested theanine under confrontational housing may be explained by the changes in Kyn and carnosine. Carnosine is synthesized from β-alanine and histidine. In the SAMP10 mice, β-alanine and histidine tended to increase with theanine intake (Appendix A).

Theanine has been reported to suppress the depression-like behavior in mice and rats using the tail suspension test, forced swimming test, and elevated plus maze test [28,29,30]. Ogawa et al. [29] measured amino acid levels in cerebrospinal fluid and found that glutamate and methionine were increased in mice that ingested theanine. Shen et al. [30] measured monoamine levels in the limbic-cortical-striatal-pallidal-thalamic circuit in stressed mice. We showed the data of methionine and glutamate in the hippocampus (Appendix A), but these results could not be compared due to different experimental conditions.

Npas4 regulates the formation and maintenance of inhibitory synapses in response to excitatory synaptic activity [7,31]. On the basis of our previous data [4], we focused on the levels of excitatory and inhibitory neurotransmitters, glutamate and γ-aminobutyric acid (GABA), but significant change was not observed (Appendix A).

Histamine has strong effects on the excitability in the hippocampus, and histamine release is enhanced by a variety of stressors [32]. The synthesis of histamine is under the control of inhibitory H_3_ autoreceptors located on histamine neurons [33]. However, theanine intake significantly suppressed histamine levels in the SAMP10 mice under confrontational housing, suggesting that the histaminergic system was an important target for theanine. In addition, histamine is strongly suggested to have a pivotal role in the regulation of sleep and wakefulness via H_1_ or H_3_ receptor [33]. Theanine has been suggested to improve sleep quality based on actigraph-based sleep studies [34] and based on the Pittsburgh Sleep Quality index [35], but the mechanism remains unclear. The high histamine levels in the SAMP10 mice could not be explained by HDC expression levels, as HDC expression in the hypothalamus was higher in the ddY mice than in the group reared SAMP10 mice. Further research is needed on the effects of theanine on histaminergic nerves.

Histidinol is dehydrogenated to histidine, suggesting that elevated histidinol levels in the SAMP10 mice may be partly involved in elevated histamine levels. In addition, 2,4-diaminobutyric acid has been reported to inhibit neuronal GABA transport [36]; GABA is a main inhibitory transporter. GABA levels did not differ at all in these mice (Appendix A), but increased levels of histamine, histidinol, and 2,4-diaminobutyric acid may make the balance between excitability and inhibitory states predominantly excitatory. It has been reported that longevity is dynamically regulated by the excitatory–inhibitory balance of neural circuits [37]. Increased excitation may have contributed to the shortened lifespan of confrontational-housing SAMP10 mice [3].

5-Methoxytryptamine is an agonist of serotonin receptors. Although increased levels of 5-methoxytryptamine may enhance the stress-related adaptive behavioral responses [38], its role is currently unknown. 

Put and Orn are metabolites of arginine (Arg), a semi-essential amino acid that is metabolized to form a number of bioactive molecules such as nitric oxide (NO) [39]. Arg is hydrolyzed by arginase to Orn, and Orn becomes Put by ornithine decarboxylase. Polyamines containing Put, spermidine, and spermine are essential for normal cellular function such as neurogenesis and aging [40]. Orn was significantly increased with theanine intake in the SAMP10 mice under confrontational housing and tended to be increased under group housing. In the SAMP10 mice, it was suggested that Orn synthesis was increased by increased expression of arginase by theanine ingestion. Orn has been reported to have an antistress effect [41]. In addition, the antistress effect of Arg has been confirmed in mice [42]. While the level of Orn was lower than that of Arg in the hippocampus, the increase in Orn with theanine intake may be important.

Adenosine is present in all nervous cells containing neurons and glia, and it modulates to lead to the homeostatic coordination of brain function [43]. Adenosine activates membrane-located G-protein coupled receptors. The A1 receptor is an inhibitory receptor coupled with Gi/o proteins, and the A2A receptor is an excitatory receptor coupled with Gs proteins [44]. It has been reported that chronic stress altered adenosine metabolism in a zebrafish brain [45]. However, the expression levels of adenosine and Put were further reduced by the ingestion of theanine under confrontational-housing conditions in the SAMP10 mice. These results suggest that changes in adenosine and Put are not important in the antidepressant action of theanine in the SAMP10 mice. The levels of GMP were lower in the SAMP10 mice than in the ddY mice, but the metabolic role in stress sensitivity is currently unknown. 

Summarizing the above, when hippocampal metabolites were compared between the SAMP10 and ddY mice that were stressed for one month, Kyn and histamine were higher in the SAMP10 mice than in the ddY mice. On the one hand, their expression was suppressed in the SAMP10 mice that had ingested theanine under confrontational housing, on the other hand, the expression of carnosine increased in SAMP10 mice that had ingested theanine during stress loading. In addition, Orn was increased in the SAMP10 mice through increased expression of arginase by ingestion of theanine. These metabolic changes were well correlated with the improvement in depression-like behavior in SAMP10 mice with theanine intake under chronical stress.

Actually, theanine has been reported to have beneficial effects on depressive disorder, anxiety, sleep disorders, and cognitive decline in patients with major depressive disorder, and on stress-related symptoms and cognitive functions in healthy adults [35,46]. The results of our study are considered to be important clues for elucidating how theanine acts in the brain to ameliorate stress-related symptoms.

This study has some limitations. First, the brain metabolites examined in this study mainly focused on 60 metabolites, which were amine-mediated metabolites based on the pathophysiology of the brain in Alzheimer’s disease. The second was that the amount of neurotransmitters actually released, such as glutamate and GABA, was not always parallel to the amount of hippocampal metabolites, making it impossible to directly assess the balance between excitation and inhibition. The third was that the evaluation of the stress-loaded period was only one month based on the stress period in which the degree of brain atrophy was most prominently observed after stress loading. However, it is necessary to consider different stress loading periods in the future.

## 4. Materials and Methods

### 4.1. Animals and Theanine Preparation 

Four-week-old male SAMP10 (SAMP10-ΔSglt2) and ddY (Slc:ddY) mice were purchased from Japan SLC Co. Ltd. (Shizuoka, Japan) and kept in conventional conditions in a temperature- and humidity-controlled room with a 12–12 h light–dark cycle (light period 08.00–20.00, temperature 23 ± 1 °C, relative humidity of 55 ± 5%). Mice were fed a normal diet (CE-2; Clea Co. Ltd., Tokyo, Japan) and water ad libitum. All experimental protocols were approved by the University of Shizuoka Laboratory Animal Care Advisory Committee (approval no. 136068 and 195241) and were in accordance with the guidelines of the U.S. National Institutes of Health for the Care and Use of Laboratory Animals.

L-Theanine (suntheanine; Taiyo Kagaku Co. Ltd., Yokkaichi, Japan) was used at 20 µg/mL normal tap water, according to previous data [3,5]. Mice consumed a theanine solution ad libitum. The theanine solution was freshly prepared twice a week. The mouse intake of theanine was equivalent to 6 mg/kg.

### 4.2. Housing Conditions for Stress Experiments

Four-week-old mice were housed in groups of six per cage for five days to habituate them to novel conditions. Then, mice were divided into two groups, namely, confrontational and group housing, according to a previously described method (Figure 3) [3]. In brief, for confrontational housing, a standard polycarbonate cage was divided into two identical subunits by a stainless steel partition. Two mice were housed in the partitioned cage for one month to establish territorial consciousness (single housing). These mice were further divided into two groups that ingested theanine or control water. Then, the partition was removed to expose the mice to confrontational stress, and the two mice subsequently cohabited in the same cage for one month (confrontational housing). Mice were classified as follows: mice that ingested theanine under confrontational housing, mice that ingested control water under confrontational housing, mice that ingested theanine under group housing, and mice that ingested control water under group housing. The cages were placed in a styrofoam box (width 30 cm, length 40 cm, and height 15 cm) in order to avoid visual social contact between cages.

### 4.3. Measurement of Metabolites by Ultrahigh Liquid Chromatography-Tandem Mass Spectrometry (UHPLC-MS/MS)

The metabolite analytes and internal standards (stable isotope) were obtained from Wako Pure Chemical Co. (Osaka, Japan), Kanto Chemicals (Tokyo, Japan), Tokyo Kasei Co., Ltd. (Tokyo, Japan), Sigma-Aldrich (Buchs, Switzerland), and Cambridge Isotope Laboratories (Andover, USA). In addition, other regents, such as derivatization and mobile phase, were obtained from Wako Pure Chemical Co. (Osaka, Japan). In this study, we used an ACQUITY ultraperformance liquid chromatography system (UPLC-H class; Waters, MA, USA) coupled to a Xevo TQD triple quadrupole mass spectrometer equipped with an electrospray ionization source and positive mode. Reversed-phase analysis was performed using an ACQUITY UPLC BEH C18 column (1.7 µm, 2.1 × 150 mm; Waters) at 50 °C. An injection volume of 5 µL was used, and the total run time of analysis was 10 min using a mobile phase based on 0.1% formic acid in water and 0.1% formic acid in acetonitrile. Detailed information is shown in a previous report [47].

Mice were anesthetized with isoflurane, and blood was removed from the jugular vein. The brain was carefully dissected, and the hippocampus was immediately frozen. Three or four mice in each group were used for analysis. The brain-tissue sample (ca. 20 mg) was added to 1 mL of water/methanol (3:7, *v*/*v*) and internal standards (20 µL). The extraction solution with two zirconia beads (3.0 mm) in the tubes was homogenized for 10 min by a Shake Master and centrifuged at 15,000 rpm for 5 min at 4 °C for deproteinization. Then, the supernatant (500 µL) was transferred to 100 µL 0.1 M NaHCO_3_ (pH 9.0) and derivatized with an equal volume of 40 mM 9-fluorenylmethyl chloroform for 10 min at room temperature. Derivatization was halted by adding 1% formic acid (100 µL). Then, the solution was removed using a centrifugal evaporator, and residue was dissolved in 100 µL of 0.1% formic acid in water/acetonitrile (1:1, *v*/*v*). Solutions were vortexed, and 5 µL of each was analyzed by UHPLC–MS/MS.

### 4.4. Principal Component Analysis

Principal component analysis (PCA) was performed on the metabolites to compare the effects of theanine intake on controls under group or confrontational housing. To reduce the effects of individual variability among samples, PCA axes were estimated on a matrix of each group’s sample means and applied to all data [48].

### 4.5. Tail-Suspension Test

To investigate behavioral depression, the mice were individually suspended by their tails at a height of 30 cm using a clip for tail suspension (MSC2007; YTS Yamashita-Giken, Tokushima, Japan). Immobility behavior was observed for 15 min, as described previously [5,49]. Mice were considered to be immobile only when they hung passively and were completely motionless. The immobility time for the final five minutes was compared between SAMP10 mice under group or confrontational housing to examine the effect of theanine intake. Observation was similarly performed for five minutes in ddY mice. Since the optimal observation time differed depending on the mouse strain, the SAMP10 and ddY mice could not be compared, but the effect of housing condition on depression-like behavior within the same strain could be compared.

### 4.6. Quantitative Real-Time Reverse Transcription PCR (qRT-PCR)

The mice were anesthetized with isoflurane and blood was removed from the jugular vein. The brain was carefully dissected, and the hippocampus and hypothalamus were immediately frozen. The brain sample was homogenized, and total RNA was isolated using a purification kit (NucleoSpin^®^ RNA, 740955, TaKaRa Bio Inc., Shiga, Japan), in accordance with the manufacturer’s protocol. The obtained RNA was converted to cDNA using the PrimeScript^®^ RT Master Mix kit (RR036A, Takara Bio Inc., Shiga, Japan). Real-time quantitative RT-PCR analysis was performed using the PowerUp™ SYBR™ Green Master Mix (A25742, Applied Biosystems Japan Ltd., Tokyo, Japan) and automated sequence detection systems (StepOne, Applied Biosystems Japan Ltd., Tokyo, Japan). Relative gene expression was measured by previously validated primers for IDO and TDO [50], HDC [51], and arginase [52] genes. The primer sequences were mentioned in Table 3. cDNA derived from transcripts encoding β-actin was used as the internal control.

### 4.7. Statistical Analyses

With SAMP10 or ddY, only substances with a *p*-value less than 0.05 in the housing condition or theanine treatment were selected and applied to PCA to reduce experimental noise. To cancel individual differences in the samples, the axes were calculated from the mean for each group and applied to all data [53]. Each substance log data were centered prior to PCA. Values of PCs were scaled [54]. Confidence intervals and significance of differences in means were estimated by using Fisher’s least significant difference test. 

## 5. Conclusions

Increased Kyn and decreased carnosine levels are associated with depression-like behavior in SAMP10 mice. An increased histamine level may be a reason for the shortened lifespan of stress loaded SAMP10 mice. In addition, an increase in Orn due to theanine intake may have a role in stress reduction. Theanine was indicated to reduce stress vulnerability by correcting those metabolic alterations.

## Figures and Tables

**Figure 1 ijms-22-00193-f001:**
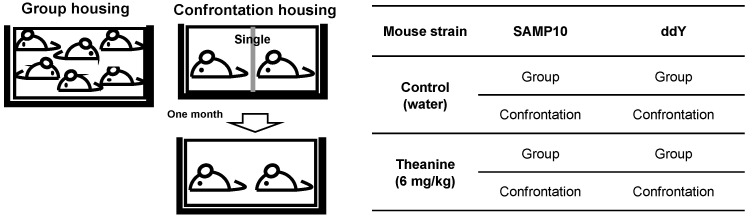
Experimental protocol.

**Figure 2 ijms-22-00193-f002:**
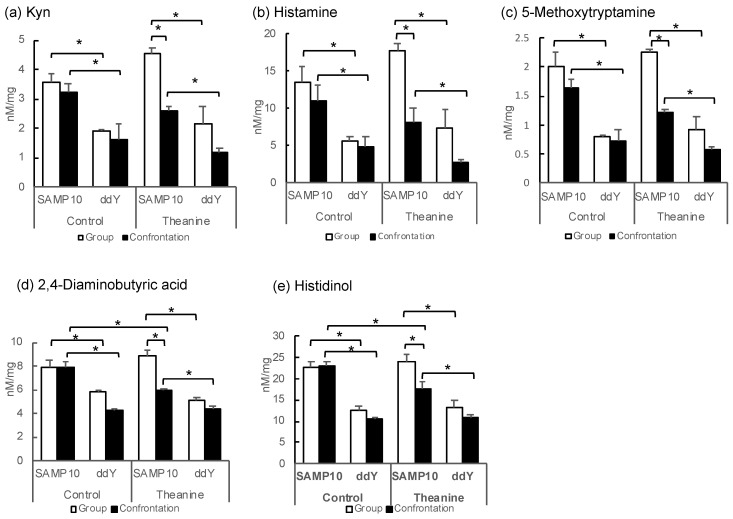
Metabolite levels in the hippocampus of SAMP10 and ddY mice; (**a**) Kyn, (**b**) histamine, (**c**) 5-methoxytryptamine, (**d**) 2,4-diaminobutyric acid, (**e**) histidinol, (**f**) GMP, (**g**) carnosine, (**h**) Put, (**i**) Orn, and (**j**) adenosine. Mice were housed confrontationally for one month after single housing for one month (closed column). Group-housed mice were kept in a group of six for two months (open column). Mice ingested theanine (20 µg/mL, 6 mg/kg) or water (control) for two months (*n* = 3–4, *, *p* < 0.05).

**Figure 3 ijms-22-00193-f003:**
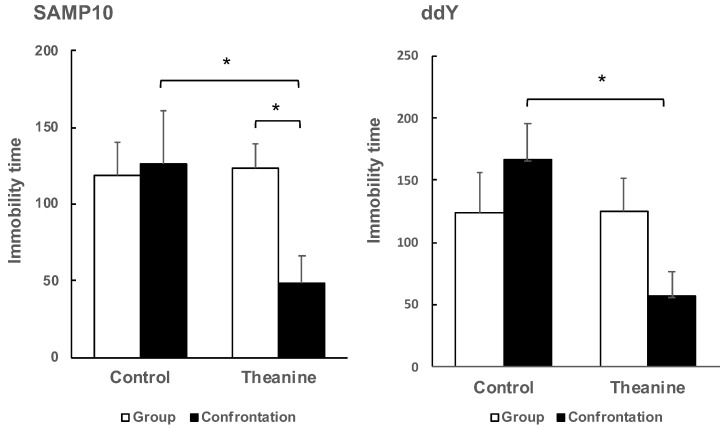
Effect of theanine intake on depression-like behavior of SAMP10 and ddY under group or confrontational housing. Mice were housed confrontationally for one month after single housing for one month (closed column). Group-housed mice were kept in a group of four for two months (open column). Mice ingested theanine (20 µg/mL, 6 mg/kg) or water (control) for two months (*n* = 4, *, *p* < 0.05).

**Figure 4 ijms-22-00193-f004:**
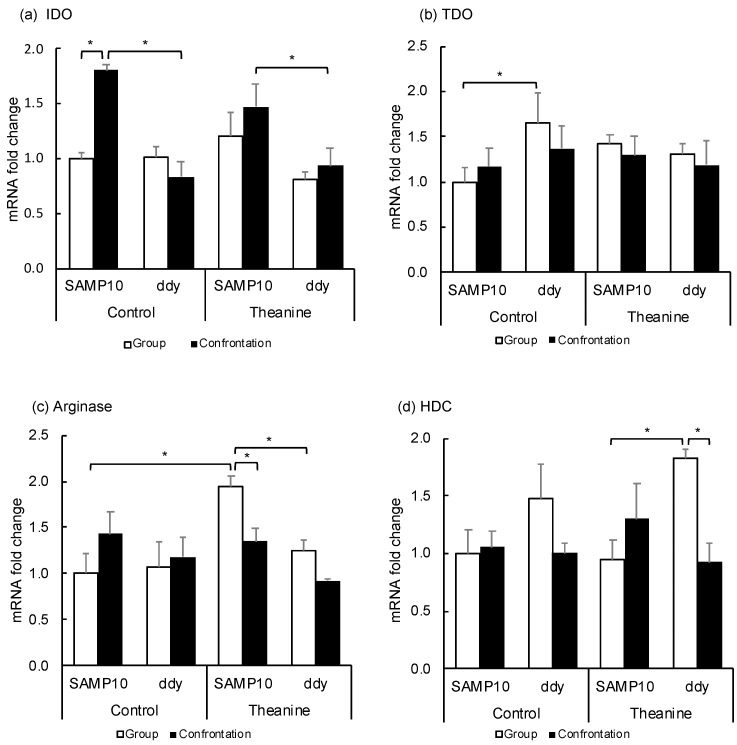
Effect of theanine intake on mRNA expression of SAMP10 and ddY mice under group or confrontational housing. (**a**) Indoleamine-2,3-dioxygenase (IDO); (**b**) Tryptophan-2,3-dioxygenase (TDO); (**c**) Arginase; (**d**) Histidine decarboxylase (HDC). Mice were housed confrontationally for one month after single housing for one month (closed column). Group-housed mice were kept in a group of four for two months (open column). Mice ingested theanine (20 µg/mL, 6 mg/kg) or water (control) for two months (*n* = 4, *, *p* < 0.05).

**Table 1 ijms-22-00193-t001:** Data of principal component analysis of metabolites in senescence-accelerated mouse prone 10 (SAMP10) and ddY mice.

Mouse	Condition	Treatment	PC1	PC2	PC3	PC4
SAMP10	Group	Control	–0.07338	0.03785	–0.00872	–0.00107
			–0.03254	–0.00421	–0.02051	–0.00247
			–0.07107	0.03476	–0.00459	–0.00835
			–0.03904	–0.00196	–0.00954	–0.01278
		Theanine	–0.07666	0.02389	0.00652	0.01317
			–0.06840	0.02850	–0.00088	0.00442
			–0.07974	0.02936	0.01287	0.00789
	Confrontation	Control	–0.04038	0.00882	0.01114	–0.02924
			–0.04848	–0.01130	0.00611	–0.01154
			–0.06143	0.00294	0.01720	0.00465
			–0.00513	–0.06122	0.00394	0.00318
		Theanine	0.00952	–0.10804	–0.01076	0.00681
			–0.00350	–0.05210	0.01452	0.02649
			0.01278	–0.11432	–0.00996	–0.01817
			–0.02355	–0.03912	0.00415	–0.00177
ddY	Group	Control	0.01670	0.00225	–0.01376	–0.02015
			0.01147	0.00520	–0.01819	–0.00116
			0.02528	0.00516	–0.01773	0.01430
		Theanine	–0.00682	0.00349	0.00519	0.01326
			–0.04108	–0.01144	–0.00352	0.02319
			0.02797	0.00704	0.00463	0.00567
			0.17347	0.04502	0.02486	0.01496
	Confrontation	Control	0.09426	0.02753	–0.00925	0.01045
			0.04019	0.00548	–0.02618	–0.01805
			0.09584	0.02586	0.01472	0.00620
			–0.01268	–0.03017	–0.02188	0.02948
		Theanine	0.07307	0.01096	0.02558	–0.03502
			0.06694	0.01848	0.00251	0.00074
			0.04413	0.00230	–0.02052	–0.02141
			0.14107	0.03713	0.02662	0.01047

Note, mice were housed in groups of six per cage. After establishing territorial consciousness by single housing for one month, then, confrontational housing was carried out for one month. These mice ingested theanine or control water. The color gradient in each column indicates the level of metabolites. The darker the red, the higher than the overall average, and the darker the blue the lower than the overall average.

**Table 2 ijms-22-00193-t002:** Statistically different metabolites in the hippocampus of SAMP10 and ddY mice.

Metaborites	PC1	PC2
Kynurenine	−0.12390	0.02227
Histamine	−0.06985	0.00491
5-Methoxytryptamine	−0.03622	−0.08937
2.4-Diaminobutyric.acid	−0.03085	−0.00003
Histidinol	−0.02561	0.00316
5-Aminovaleric.acid	−0.02370	−0.00610
Cadaverine	−0.02067	−0.00384
Diacetyl spermidine	−0.01859	0.00364
3-Methoxyanthranilate	−0.01750	−0.00783
Leucine	−0.00513	0.00335
Methionine	−0.00364	0.00280
Proline	−0.00277	0.00327
Creatinine	−0.00269	−0.00184
2-Aminoadipate	−0.00263	−0.00433
Aspartic acid	−0.00229	−0.00076
Arginine	−0.00213	0.00345
Cystine	−0.00201	0.00122
2-Aminobutyric.acid	−0.00160	0.00033
Valine	−0.00124	0.00227
Phenylalanine	−0.00119	0.00302
Alanine	−0.00084	0.00084
β-Alanine	−0.00064	0.00208
N^G^-monomethyl-arginine	−0.00028	−0.00849
Isoleucine	−0.00019	0.00310
Glutamine	−0.00008	−0.00027
Histidine	0.00037	0.00187
Hypoxanthine	0.00103	0.00440
Glutamic acid	0.00132	−0.00123
Spermidine	0.00143	−0.00015
Glutathione reduced	0.00146	−0.00238
Serine	0.00162	−0.00050
CMP	0.00262	0.00253
Adenine	0.00368	−0.01675
Adenosine	0.00482	−0.00904
Ornithine	0.00487	−0.00004
Putrescine	0.00668	−0.01895
Carnosine	0.00901	0.00522
GMP	0.01038	−0.00718

GMP, guanosine monophosphate and CMP, cytidine monophosphate. The color gradient in each column indicates the level of metabolites. The darker the red, the higher than the overall average, and the darker the blue the lower than the overall average.

**Table 3 ijms-22-00193-t003:** Sequence of primers used in qRT-PCR analysis.

Gene	Forward Sequence	Reverse Sequence	Ref.
IDO	GGGCTTCTTCCTCGTCTCTC	TGGATACAGTGGGGATTGCT	[50]
TDO	TCCAGGGAGCACTGATGATA	CTGGAAAGGGACCTGGAATC	[50]
HDC	CGTGAATACTACCGAGCTAGAGG	ACTCGTTCAATGTCCCCAAAG	[51]
Arginase	CATGGGCAACCTGTGTCCTT	TCCTGGTACATCTGGGAACTTTC	[52]

## Data Availability

Not applicable.

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
