# Peer review of "Theanine, Antistress Amino Acid in Tea Leaves, Causes Hippocampal Metabolic Changes and Antidepressant Effects in Stress-Loaded Mice"

_ijms, 2020, doi:10.3390/ijms22010193_

Round 1

Reviewer 1 Report

The work entitled „Theanine, Antistress Amino Acid in Tea Leaves,  Causes Hippocampal Metabolic Changes and Antidepressant Effects in Stress-Loaded Mice” is interesting and subject of the research may bring some important new data on how stress triggers depression-like behaviors and the effects of theanine ingestion as an anti-stress agent.

This study, however, should be refined and, in my opinion, lacks the full molecular profiling.

  • It would be beneficiary to the manuscript if the author add figure (or move Figure 3 to the results section) that explains the confrontational housing experiment.
  • Some abbreviations are not explained, like: GMP, CMP, ddY
  • Figure 2 results and methods of the tail-suspension test should be better explained since it is the only behavioral test used in the study.
  • Why the tail-suspension test was not performed for ddY mice?
  • What is greatly missing in this study is molecular aspect of the difference: like showing the molecular pathways differences (i.e. RNA/protein expression analysis) between particular study groups in brain tissues (hippocampus), like enzymes involved in synthesis and metabolism of the hippocampal metabolites - since the brain tissue was already isolated. What is more, there is no discussion on this subject either.  
  • The Discussion section is also missing two important parts, first, there is no reference to other studies where theanine was used and this behavioral test was used – are the results similar? There is a description of similar experiment (or maybe the same experiment) in the introduction part of the manuscript (ref [4]) – I guess it should be also discussed in the Discussion. Second, how this results may translate into human physiology?
  • What are the limitations of the study?

Author Response

The work entitled „Theanine, Antistress Amino Acid in Tea Leaves,  Causes Hippocampal Metabolic Changes and Antidepressant Effects in Stress-Loaded Mice” is interesting and subject of the research may bring some important new data on how stress triggers depression-like behaviors and the effects of theanine ingestion as an anti-stress agent.

This study, however, should be refined and, in my opinion, lacks the full molecular profiling.

Thank you very much for reviewing our manuscript. Our manuscript has been revised according to the helpful opinions of the reviewer. In addition, we have revised some of the explanations in Tables 1 and 2 in the results (Section 2.1).

  • It would be beneficiary to the manuscript if the author add figure (or move Figure 3 to the results section) that explains the confrontational housing experiment.

The experiment of confrontational housing and Figure 3 was moved to the results.

  • Some abbreviations are not explained, like: GMP, CMP, ddY

Thanks. These abbreviations were explained, but “ddY” is the official mouse strain name. 

  • Figure 2 results and methods of the tail-suspension test should be better explained since it is the only behavioral test used in the study.

The explanation of result and method of Figure 2 (Figure 3, in the revised manuscript) were added that are shown in red characters.

  • Why the tail-suspension test was not performed for ddY mice?

Since the focus of this paper was on SAMP10, only the data for SAMP10 was shown.

As similar result was obtained for ddY, the results were added.

  • What is greatly missing in this study is molecular aspect of the difference: like showing the molecular pathways differences (i.e. RNA/protein expression analysis) between particular study groups in brain tissues (hippocampus), like enzymes involved in synthesis and metabolism of the hippocampal metabolites - since the brain tissue was already isolated. What is more, there is no discussion on this subject either.  

Thank you for your valuable suggestion.We examined changes in RNA expression of major enzymes such as the levels of indoleamine/tryptophan-2,3-dioxygenase (IOD/TOD), arginase and histidine decarboxylase (HDC). And added the data as new results, and considered their involvement in the discussion.

  1. The expression of IOD and TOD enzymes involved in Trp to Kyn metabolism, in the hippocampus.
  2. The expression of histidine decarboxylase (HDC) involved in histamine synthesis from histidine in the hypothalamus.
  3. The expression of arginase involved in ornithine synthesis from arginine in the hippocampus.
  • The Discussion section is also missing two important parts, first, there is no reference to other studies where theanine was used and this behavioral test was used – are the results similar? There is a description of similar experiment (or maybe the same experiment) in the introduction part of the manuscript (ref [4]) – I guess it should be also discussed in the Discussion. Second, how this result may translate into human physiology?

Some studies containing our data (ref. 4) about theanine were discussed.

We discussed the possibility of our data to translate into human physiology.

  • What are the limitations of the study?

The limitations of our study were added in the discussion.

Reviewer 2 Report

Very interesting results pointing to the potential role of theanine in antidepressant activity.
The authors used SAMP10 mice and investigated the levels of metabolites essential for stress and depression in the hippocampus.
The use of SAMP10 mice in research may rather suggest that observed data are as a result of aging brain. Are there any data correlating the obtained metabolite results with this process?
Authors should use two-way ANOVA (strain x theanine) etc.
Why do the authors use the tail suspension test for antidepressant activity and not the Swim test?

Author Response

Very interesting results pointing to the potential role of theanine in antidepressant activity.
The authors used SAMP10 mice and investigated the levels of metabolites essential for stress and depression in the hippocampus.
The use of SAMP10 mice in research may rather suggest that observed data are as a result of aging brain. Are there any data correlating the obtained metabolite results with this process?
Thank you very much for reviewing our manuscript.

As the reviewer pointed out, stress loading may have accelerated aging, but it cannot be clarified now. In the near future, it will be necessary to study age-related changes in brain metabolites.

Authors should use two-way ANOVA (strain x theanine) etc.

It was found that brain atrophy occurs with confrontational housing for one month in both SAMP10 and ddY, but since there was a difference in the degree of atrophy due to theanine intake, this study attempted one-way ANOVA without considering the difference in mouse strains.

In this revision, we have revised some of the explanations in Tables 1 and 2 of the results (Section 2.1) and the explanations for statistical analysis (Section 4.7).Why do the authors use the tail suspension test for antidepressant activity and not the Swim test?

Forced swim test can be more stressful to mice than tail suspension test, as mice generally hate getting wet. To minimize the effect on brain metabolites, only the tail suspension test was performed.

Round 2

Reviewer 1 Report

In general I accept the changes introduced by authors in response to my comments.

There is only one thing missing - Table 3 with primer sequences.

Author Response

In general I accept the changes introduced by authors in response to my comments.

There is only one thing missing - Table 3 with primer sequences.

Thank you very much for reviewing our revised manuscript and pointing out our mistake.

We added Table 3.